# Multi-UAV Collaborative Search and Attack Mission Decision-Making in Unknown Environments

**DOI:** 10.3390/s23177398

**Published:** 2023-08-24

**Authors:** Zibin Liang, Qing Li, Guodong Fu

**Affiliations:** 1Beijing Key Laboratory of High Dynamic Navigation Technology, Beijing 100192, China; zibinl@163.com (Z.L.); fuguodd@163.com (G.F.); 2Ministry of Education Key Laboratory of Modern Measurement & Control Technology, Beijing 100101, China; 3School of Automation, Beijing Information Science & Technology University, Beijing 100192, China

**Keywords:** multi-UAV, coordinated operations, Q-learning, ant colony algorithm, artificial potential field

## Abstract

To address the challenge of coordinated combat involving multiple UAVs in reconnaissance and search attacks, we propose the Multi-UAV Distributed Self-Organizing Cooperative Intelligence Surveillance and Combat (CISCS) strategy. This strategy employs distributed control to overcome issues associated with centralized control and communication difficulties. Additionally, it introduces a time-constrained formation controller to address the problem of unstable multi-UAV formations and lengthy formation times. Furthermore, a multi-task allocation algorithm is designed to tackle the issue of allocating multiple tasks to individual UAVs, enabling autonomous decision-making at the local level. The distributed self-organized multi-UAV cooperative reconnaissance and combat strategy consists of three main components. Firstly, a multi-UAV finite time formation controller allows for the rapid formation of a mission-specific formation in a finite period. Secondly, a multi-task goal assignment module generates a task sequence for each UAV, utilizing an improved distributed Ant Colony Optimization (ACO) algorithm based on Q-Learning. This module also incorporates a colony disorientation strategy to expand the search range and a search transition strategy to prevent premature convergence of the algorithm. Lastly, a UAV obstacle avoidance module considers internal collisions and provides real-time obstacle avoidance paths for multiple UAVs. In the first part, we propose a formation algorithm in finite time to enable the quick formation of multiple UAVs in a three-dimensional space. In the second part, an improved distributed ACO algorithm based on Q-Learning is introduced for task allocation and generation of task sequences. This module includes a colony disorientation strategy to expand the search range and a search transition strategy to avoid premature convergence. In the third part, a multi-task target assignment module is presented to generate task sequences for each UAV, considering internal collisions. This module provides real-time obstacle avoidance paths for multiple UAVs, preventing premature convergence of the algorithm. Finally, we verify the practicality and reliability of the strategy through simulations.

## 1. Introduction

In recent years, with the increase in mission complexity and the enhancement of UAV autonomy, unmanned aircraft swarm systems (UAS) have gradually received attention from researchers. Drone swarms have been used for target search and rescue [1,2], cluster operation [3,4], and target tracking [5,6]. Within the military domain, coordinated reconnaissance search attacks by UAV swarms in unknown environments is an important application. Drone swarm reconnaissance operations can provide more information, faster response times, greater accuracy, and lower costs. UAV swarm missions are characterized by complexity and dynamics, making multi-objective tasking of multiple UAVs a challenge [7,8].

Task allocation is the assignment of a task to an intelligence or a group of intelligences, with the goal of finding an optimal or near-optimal mapping between an intelligence and a task [9]. Auction or market-based approaches are commonly used in multi-robot task allocation problems [10,11], including AsyMTRE-D [12], Murdoch [13], M+ [14], and TraderBots [15]. Graph-theoretic particle swarm optimization for military applications [16], Sandholm’s algorithm with K-means clustering [17], Efficiency-based tasking [18], and many other approaches focus on minimizing the distance carried out by the robot to complete the task while satisfying the robot’s own resource allocation. Borja Fernandez-Gauna et al. developed a distributed round-robin Q-Learning for the transportation of hoses using UGVs [19]. Mathew et al. developed path planning algorithms for miniature UAVs and UGVs performing parcel delivery tasks [20]; the problem is modeled as a multi-warehouse delivery problem, transformed into a generalized traveler’s problem, and solved in polynomial time using a kernel sequence enumeration algorithm. Bayesian Reinforcement Learning (RL) allows the agents to learn the capabilities of other agents through interactions and transform repetitive coalition formation problems into sequential decision problems [21]. This approach was validated in the soccer team formation problem [22], Modeling Dynamic Robot Formation for the Area Coverage Problem Using Weighted Voting Games and Q-Learning, and Extended to Formation-Based Navigation Problems [23,24]. The formation structure is pruned using Shapley values and marginal contributions, and the transition of robots from one formation to another is represented by the Markov process, which searches for the optimal structure of the formation space using a Markov probability distribution. Sayan D. et al. proposed a multi-criteria decision-making algorithm based on influence diagrams to select the best formation algorithm for a given realistic scenario [25]. Pengxing Zhu et al. constructed a heterogeneous UAV multi-task assignment model based on multiple realistic constraints and proposed an improved semi-randomized Q-Learning algorithm, which was demonstrated through simulation to be able to improve the performance of task execution compared to heuristic algorithms such as the Q-Learning algorithm, including improving the reasonableness of the task assignment, with a gain value of improvement of 12.12% [26]. Han Qi et al. solved the target assignment and trajectory planning separately and proposed a multi-intelligence target assignment and trajectory planning method based on a multi-intelligence reinforcement learning algorithm [27]. Minchan Li et al. proposed an effective stability quantum particle swarm optimization algorithm (SQPSO) and designed an efficient coalition-building quantum particle swarm optimization algorithm (EQPSO), which incorporates coalition similarity judgments and reduces coalition formation time overheads [28]. Heba Kurdi et al. proposed a bacterial heuristic algorithm for UAV task assignment [29]. Xiang Ma et al. proposed an improved virtual force method to obtain the optimal damage effectiveness control strategy, which solves the numerical solution problem caused by the high complexity of the damage function [30]. Xia Chen et al. proposed a systematic framework for solving the combinatorial optimization model with a new particle swarm optimization algorithm based on a bootstrapping mechanism, and for the trajectory planning problem, an ant colony optimization algorithm based on adaptive parameter tuning and bidirectional search was proposed [31]. Taehoon Yoo et al. proposed a reinforcement learning-based intelligent formation controller that enables each UAV to communicate with other UAVs [32]. Yuchong Gao et al. proposed a scalable, purely azimuthal passive UAV formation keeping distributed control algorithm to maintain multi-UAV formation using pure angular information through only a small amount of inter-UAV communication [33].

In this paper, a multi-UAV-distributed self-organized collaborative intelligent reconnaissance operation strategy is proposed for the problem of collaborative reconnaissance operation of multi-UAV multi-mission targets. We have performed the following main tasks:A distributed self-organized multi-UAV cooperative intelligent reconnaissance combat strategy (CISCS) is proposed for the multi-UAV cooperative reconnaissance combat problem;A formation controller with time constraints is proposed to extend the multi-layer artificial potential field to avoid the internal collision of multiple UAVs in fast formation and to solve the problem where the traditional artificial potential field method tends to fall into the optimal solution;Reinforcement learning is utilized to accelerate the convergence speed of the ACO algorithm and expand the search capability, and a colony disorientation strategy and a search-switching strategy are incorporated to prevent its premature maturity.

We will discuss the above according to the following structure: Section 2 is the multi UAV multi-task target feature scene and task allocation decision model; Section 3 designs the overall structure of CISCS, including a formation controller with time constraints for fast formation, a distributed and improved ant colony algorithm based on Q-Learning for generating multi-UAV task sequences, and an extended artificial-potential field algorithm for drone avoidance within the drones formation; in Section 4, we conducted simulation and demonstration of this algorithm; and the conclusions are presented in Section 5.

## 2. Scene Description and Modeling

In this section, the search attack task in an uncertain environment is modeled; the search attack task planning problem is defined, and the constraints of the search attack task planning model are given.

**Assumption** **1.**
*Each UAV is capable of performing reconnaissance search and attack missions.*


$V=\{V_{1},V_{2},\ldots ,V_{N_{v}}\}$ indicates $N_{v}$ multi-drone formation. $T=\{T_{1},T_{2},\ldots ,T_{N_{t}}\}$ is the set of $N_{t}$ unknown targets distributed within the task area.

### 2.1. Collaborative Search Attack Constraints

Multi-UAV coordinated search–attack mission is an online planning problem whose goal is to discover and destroy as many targets as possible in a dynamic and complex environment. Therefore, the primary problem is to develop a distributed optimization model for search–attack missions. A detailed description of these constraints is given below [34]:(1)Maximum range constraint

Single UAV mission with maximum range Dmax:
*D*_max_ − *D_i_*(*t*) ≥ 0 (*i* = 1, 2, 3, …, *N_v_*)(1)
where *D_i_*(*t*) denotes the cumulative distance traveled by the *i*th UAV before moment *t*; *D_i_*(*t*) denotes the maximum distance traveled;
(2)Maximum turning angle constraint

The drone has a maximum turning angle $\theta_{max}$
(2)$$\theta_{max}-\theta_{i}(t)\geq 0\;(i=1,\;2,\;3,\;\ldots ,\;N_{v})$$
where $\theta_{i}(t)$ denotes the turning angle of the *i*th UAV at moment *t*. The maximum turn angle limits the UAV turn radius and turn rate, $\theta_{max}\in \;(0,\;\pi )$. When the desired turn angle is 0, it means that the drone does not need to turn; when the desired turn angle is pi, it means that the drone is expected to fly in the reverse direction;
(3)Minimum flight distance between drones

In order to ensure the flight safety of multi-UAV formations, there exists a safety distance for multi-UAVs to perform their tasks:
(3)$$\;d_{ij}\left(t\right)-d_{min}\geq 0\;(i=1,\;2,\;3,\;\ldots ,\;N_{v})$$
where $d_{ij}\left(t\right)$ denotes the distance between the *i*th UAV and the *j*th UAV at time *t*; $d_{min}$ denotes the minimum flight distance between any two drones;
(4)Risk area avoidance constraints

UAVs need to maneuver to avoid the risk zone during combat, so there is a minimum risk radius between the UAV and the center of the risk zone:
(4)$$d_{il}\left(t\right)-R_{min}\geq 0\;(i=1,\;2,\;3,\;\ldots ,\;N_{v},\;l=1,\;2,\;3,\;\ldots ,\;N_{l})$$
where *d_il_*(*t*) denotes the distance between the *i*th UAV and the *l*th risk zone at time *t*; *R*_min_ denotes the minimum risk radius.

### 2.2. Multi-UAV Collaborative Mission Description

(1)Description of reconnaissance coverage

Ground reconnaissance coverage radius exists for UAVs during reconnaissance missions.

Reconnaissance coverage:(5)$$P_{r}=\sum_{i=1}^{N_{v}}\frac{S_{i}}{S_{Area}}$$
where *S_i_* denotes the reconnaissance area of the *i*th UAV; *S_Area_* denotes the total area of the combat zone.

Reconnaissance search benefit function:*J_r_* = *P_r_*(6)

(2)Description of the attack mission

Attack gain function:(7)$$J_{a}=\sum_{i=1}^{N_{v}}\sum_{j=1}^{T}a_{ij}{score}_{j}$$
where *a_ij_* denotes that the *i*th UAV attacked the *j*th target; when *a_ij_* = 0, it means that the attack mission on the *j*th target was not accomplished; when *a_ij_* = 1, it means that the attack mission on the *j*th target was accomplished. *score_j_* denotes the gain of the attack on the *j*th target.

### 2.3. Mathematical Model for Tasking

The coordinated multi-UAV search attack needs to maximize the benefit function under the above constraints, so the following centralized multi-UAV task allocation constraint model can be established:(8)$$U^{*}=arg\;\underset{U}{\max}\left(\eta J_{r}+(1-\eta )J_{a}\right)\;s.t.\left\{\begin{array}{c} D_{max}-D_{i}\left(t\right)\geq 0\\ \begin{array}{c} \theta_{max}-\theta_{i}\left(t\right)\geq 0\\ d_{ij}\left(t\right)-d_{min}\geq 0\\ d_{il}\left(t\right)-R_{min}\geq 0 \end{array},\;(i,j=1,2,\ldots ,N_{v},i\neq j) \end{array}\right.$$
where *η* ∈ {0, 1}; when *η* = 0, it denotes that the UAV performs an attack mission; when *η* = 1, it denotes that the UAV performs a reconnaissance mission. *U* denotes the UAV decision-making. Assuming that each UAV is an independent individual and all have the ability to make independent decisions, the centralized optimization model (8) can be decomposed into a distributed optimization model:(9)$$U_{i}^{*}=arg\;\underset{U_{i}}{\max}\left(\eta_{i}J_{ri}(X_{i},{\tilde{X}}_{j})+(1-\eta_{i})J_{ai}(X_{i},{\tilde{X}}_{j})\right)\;s.t.\left\{\begin{array}{c} D_{i\;max}-D_{i}\left(t\right)\geq 0\\ \begin{array}{c} \theta_{i\;max}-\theta_{i}\left(t\right)\geq 0\\ d_{ij}\left(t\right)-d_{ij\;min}\geq 0\\ d_{il}\left(t\right)-R_{i\;min}\geq 0 \end{array},\;(i,j=1,2,\ldots ,N_{v},i\neq j) \end{array}\right.$$
where $U_{i}^{*}$ denotes the decision of the *i*th drone, $J_{ri}$ denotes the reconnaissance gain of the *i*th UAV, $J_{ai}$ denotes the gain from the *i*th drone attack; *X_i_* denotes the state of the *i*th drone; ${\tilde{X}}_{j}=\{X_{j}|j\in T_{adjacent}^{i}\}$, where $T_{adjacent}^{i}$ denotes the set of UAVs that can communicate with the *i*th UAV.

## 3. Algorithms for Multi-Drone Collaborative Intelligent Warfare

Based on the above-distributed optimization model (9), multi-UAVs interact with the battlefield for information, real-time formation control of multi-UAVs, task assignment, and path planning. In this section, we will design a multi-UAV cooperative intelligent combat algorithm for the multi-UAV real-time target search and attack problem.

### 3.1. Formation Controller with Time Constraints

When multiple UAVs fly in formation, the Leader sends position information to the follower, and the follower adjusts its desired position coordinates in real time according to the Leader’s position information and, eventually, forms into a stable formation [35]. In order to improve the rapidity of multi-UAV formation, we propose a finite time-based multi-UAV formation control algorithm, as shown in Equation (10).
(10)$$u_{i}^{f}(t)=k_{f}[P\left(\theta_{i}\left(t\right),\;\varphi_{i}\left(t\right)\right)\left(d_{f}-d_{i}\left(t\right)\right)+k_{i}\int_{t_{0}}^{t_{1}}\left(d_{f}-d\left(t\right)\right)dt\;+k_{d}\frac{d\left(d_{f}-d\left(t\right)\right)}{dt}]$$
(11)$$k_{f}=\frac{w}{t_{w}-t},\;t\in [0,\;t_{w})$$

In Equation (10), $u_{i}^{f}\left(t\right)$ is the input of UAV *i*; $k_{f}$ is the finite time gain variable; $t_{w}$ indicates limited time; $d_{f}$ is the distance between the drone and the formation target; $d_{f}$ is the desired distance; $d_{i}\left(t\right)$ is the distance between UAV *i* and the formation target at the moment *t*; $\theta_{i}\left(t\right)$ is the angle between the line of the leader and the follower and the z-axis positive direction, and $\varphi_{i}\left(t\right)$ is the angle between the projection of this line on the XOY plane and the x-axis positive direction. P(θ,φ) is $\left[\begin{array}{c} sin\theta cos\varphi \\ sin\varphi cos\theta \\ cos\theta \end{array}\right]$, which is used to convert $u_{i}^{f}(t)$ to an output quantity in the x, y, and z directions.

In Equation (11), *w* is the finite time gain normal, and *t_w_* is the finite time parameter.

**Theorem** **1.***For a multi-UAV system, the finite time gain k_f_ in Equation (11) satisfies that w is a positive constant and *$t_{w}$*is an arbitrarily given finite time; then, it is guaranteed that the multi-UAVs form a leader–follower flight formation at any time t_w_*.

**Proof.** The Lyapunov function is now chosen as


(12)
$$V\left(d_{i}\left(t\right)\right)=\frac{1}{2}{\left(d_{f}-d_{i}\left(t\right)\right)}^{2}$$


Derivation leads to
(13)$$\dot{V}\left(d_{i}\left(t\right)\right)=-(d_{f}-d_{i}\left(t\right))\dot{d_{i}}\left(t\right)$$

From the controller Equation (10)
(14)$$\dot{V}\left(d_{i}\left(t\right)\right)=-(d_{f}-d_{i}\left(t\right))\cdot \frac{w}{t_{w}-t}[\left(P+k_{i}+k_{d}\right)\cdot d_{f}-P\cdot d_{i}\left(t\right)-k_{i}\int_{0}^{t_{1}}d_{i}\left(t\right)dt\;-k_{d}\cdot \frac{d}{dt}d_{i}(t)]\leq 0$$

Therefore, according to Lyapunov stability analysis, the given UAV distance controller is stable and the actual flight distance $d_{i}\left(t\right)$ converges to the desired flight distance $d_{f}$. This proves the convergence and stability of the controller. □

### 3.2. Reconnaissance Search Attack Tasking Algorithm

We propose an improved ant colony multi-task multi-objective solving method based on Q-Learning with constraints by utilizing the powerful feedback memory capability of Q-Learning [36], and the process of using this algorithm for UAV cooperation is described in detail below.

#### 3.2.1. Q-Learning State Space

The ant colony position information is used as the state space of Q-Learning, which guides the later ant colonies to find the best solution through the historical optimal path and accelerates the convergence of the ant colony algorithm. The Euclidean distance between the current state *x_t_* of the ant colony and the optimal solution of the previous state is taken as the current state *S_t_* of Q-Learning, i.e.,
(15)$$S_{t}=\left\|x_{t}-x_{t-1}\right\|$$
where $x_{t-1}$ denotes the optimal position at moment *t* − 1.

Normalize *S_t_*
(16)$$S_{t}=\left\{\begin{array}{c} HH,\;0.8\leq S_{t}\\ HM,\;0.6\leq S_{t}<0.8\\ \begin{array}{c} MM,\;0.4\leq S_{t}<0.6\\ LM,\;0.2\leq S_{t}<0.4\\ LL,\;0\leq S_{t}<0.2 \end{array} \end{array}\right.$$
where HH, HM, MM, LM, LL denote five different ant colony states, respectively.

#### 3.2.2. Q-Learning Action Space

Since ant colony search mainly involves three different parameters [37]: pheromone-inspired factor α; expectation-inspired factor β; and pheromone evaporation factor ρ. In this paper, we adaptively adjust the size of the three parameters according to the environment and regulate the search capability into five grades, namely, full-domain search, local search, hold search, slow-step convergence, and fast convergence, which are used as the action space of Q-Learning. These five search modes correspond to preset parameter combinations, as shown in Table 1.

#### 3.2.3. Reward Function

In order to speed up the process of ant colony optimization, setting reasonable rewards for the ant colony can guide the ant colony to search in a more optimal direction. Considering multiple objectives in a multi-objective optimization problem, the sum of the fitness differences before and after the ant colony update is taken as the reward value, i.e.,: (17)$$r_{t,\;t+1}=\sum_{i=1}^{n}{Fit}_{i}\left(x_{t}\right)-{Fit}_{i}(x_{t+1})$$
where $r_{t,t+1}$ denotes the reward value of the ant colony after updating; $x_{t}$ and $x_{t+1}$ denote the positions of the colony before and after updating; *n* denotes the number of objective functions; and ${Fit}_{i}\left(x_{t}\right)$ denotes the fitness of the *i*th objective function at $x_{t}$.

#### 3.2.4. Q-Value Update

Learn the design of states, actions, and rewards based on the above Q. By judging the current state the colony is in, the action corresponding to the maximum Q value of that state in the Q table is selected [38]. The combination of parameters is selected based on the optimization of this action; the state information of the ant colony is updated by the new parameters; the rewards obtained from the updating process are calculated, and the Q-table is updated by combining the new states and actions of the colony in order to track, learn, and dynamically adjust the optimization process [39].

The update function for the Q-value is:(18)$$Q\left(S_{t+1},a_{t+1}\right)=\left(1-\alpha \right)Q\left(S_{t},a_{t}\right)+\alpha [r_{t,t+1}+\gamma \underset{\alpha }{\max}Q(S_{t+1},\;a_{t})]$$
where α is the learning rate; γ is the discount; $a_{t}$ denotes the taken action at moment *t*, i.e., the update parameter at moment *t*; $r_{t,t+1}$ denotes the reward value after the update of the ant colony, and $Q\left(S_{t},a_{t}\right)$ denotes the action value of state $S_{t}$ taking action.

The algorithm flow of Q-learning is shown in Figure 1.

#### 3.2.5. Ant Colony Lost

In order to prevent the ant colony algorithm from maturing prematurely and entering the local optimal solution, this paper proposes an ant colony lost strategy to increase the local search ability of the ant colony. As shown in Figure 2.When the ant colony selects the next node according to the transfer probability, it mimics the situation of ants becoming lost and sets the lost rate τ, so that the ants do not arrive at the next node according to the agreement but be lost to another nearby node, thus increasing the local search ability of the ant colony.
(19)$${Pos}_{next}=\left\{\begin{array}{c} {Pos}_{i},\;rand\leq \tau \\ {Pos}_{j},\;else\;\;\;\;\;\;\;\;\;\; \end{array}\right.$$

It is now assumed that when the ant colony has entered into a locally optimal solution, but the colony is still in motion, there is a certain probability of the ant colony in the process of motion, i.e., the rate of disorientation; the ant colony does not follow the originally established route but is disoriented into another path, such as that of the ants in the blue circle in Figure 2, and enters into the other path, and if this disorientation obtains a new solution that is more superior to the previous one, then it means that the ant colony jumps out of the locally optimal solution.

#### 3.2.6. Search Conversion Strategy

In the multi-task objective search optimization algorithm, the ant colony is prone to premature maturity and local optimization, which makes the ant colony algorithm unable to obtain the optimal solution and lose some of the optimization search samples in order to increase the diversity of sample search; this paper designs a multi-objective search transformation strategy, which is used to enhance the algorithm’s optimization searching ability, and balances the algorithm’s ability to explore globally and develop locally. On the basis of multi-task objective ant colony optimization, the variation strategy of jumping out of the local optimum is set for the ant colony. Firstly, the ant colony enters the global optimization session to obtain the short-lived optimal solution set G. Then, it enters the local search stage and performs random mutation based on Cauchy distribution on all the solutions in G and obtains the position of the ant colony after mutation. Finally, the newly generated ant colony is evaluated for fitness and ranked in comparison with the original ant colony in G, thus updating the G table. The two search methods are used to switch continuously during the optimization process to update the optimal set G [30].

In order to fully increase the diversity of search samples, this paper adopts a variant based on the Cauchy distribution, which is used to increase the local search capability of the ant colony. Cauchy distribution is suitable for exploring randomness.

Therefore, using the nature of the Cauchy distribution to combine it with the ant colony transfer probability function, the Cauchy operator is introduced into the ant colony transfer so that the optimal solution set is updated through the Cauchy distribution-based variation to improve the local exploration ability and increase the diversity of samples. The search transformation strategy improves the searchability of the ant colony and leads the ant colony to reject the local optimal solution with a certain probability, and its formula for the ant colony variation based on the Cauchy distribution is as follows:(20)$$p_{ij}^{k}(t)\left\{\begin{array}{c} \frac{\tau_{ij}^{\alpha }\cdot \eta_{ij}^{\beta }}{\sum_{k\in C_{i}}\tau_{ij}^{\alpha }\cdot \eta_{ij}^{\beta }}+\frac{1}{2}+\frac{1}{\pi }\tan^{-1}\left(\frac{t-\tau_{ij}}{p_{ij}}\right),\;j\in C_{i}\\ 0 \end{array}\right.$$
where i,j are the start and end points, respectively; α, β are the weighting factors; τ is the pheromone concentration; η is the visibility, which is the inverse of the distance between points i,j; and C is the set of unvisited nodes.

#### 3.2.7. Improved Ant Colony Multi-Task Objective Optimization Algorithm Based on Q-Learning

The ant colony is used as the intelligent body of Q-Learning, and the ant colony updates its own action parameters through its behavioral state, which realizes the expansion of the search range, accelerates the convergence speed of the algorithm, completes the problem of solving the multi-task objective, and overcomes the problems of the algorithm being easy to enter into the locally optimal solution and precocious maturity. The algorithm flow is shown in Algorithm 1.
**Algorithm 1:** Improved Ant Colony Multi-Task Objective Optimization Algorithm Based on Q-LearningStep 1: Initialization         Set the maximum number of iterations *t*_max_, maximum number of ants *m*_max_, α, β, ρ parameters         Initialize the total amount of pheromones, Q table, and global optimal solution set G         Initialize the ant colony and randomly allocate starting points for the ants         Set initial pheromone concentration         Set evaluation function and multi-task goal planning taskStep 2: Ant colony search         For t = 1 to *t*_max_ do           For m = 1 to *m*_max_ do                  Calculate the probability of movement for each ant                  Choose the next moving node                  Ant loss strategy, choose whether to be lost in the path with a certain probability                  Update the pheromone concentration                  Evaluate the ant colony, update the optimal solution set G                  Calculate ant reward, update the action value function           End For           Generate a transient optimal solution set G           Perform search transformation strategy, update G         End ForStep 3: Output the results         Output the global optimal solution set G

#### 3.2.8. Extended Artificial Potential Field Algorithm (EAPF)

To ensure that the flight safety of multi-UAVs in the mission avoids internal collision [40], we designed a multi-UAV drone avoidance control algorithm based on the traditional artificial potential field method. The algorithm extends the multilayer potential field function to improve the UAV avoidance performance based on the traditional artificial potential field algorithm with a gravitational potential field of
(21)$$U_{att}\left(p\right)=\left\{\begin{array}{c} \alpha_{1}(p-p_{0}{)}^{2},\;d\in (0,\;d_{0})\\ 2d_{0}\alpha_{1}\left(p-p_{0}\right),\;d\in (d_{0},+\infty ) \end{array}\right.$$
where $U_{att}\left(p\right)$ is the potential field function; $\alpha_{1}$ denotes the gravitational coefficient; p denotes the current position of the UAV; $p_{0}$ is danger zone; d is the Euclidean distance between the UAV and the target, and $d_{0}$ is the distance threshold; when the distance between the position of the UAV and the location of the target is greater than the set distance threshold $d_{0}$, the gravitational force is constant. When the distance between the UAV and the target is smaller than the set distance threshold $d_{0}$, the UAV can be considered to be moving within the influence distance of the target point.

The repulsive potential field function is:(22)$$U_{rep}\left(p\right)=\left\{\begin{array}{c} 0,\;d\in (d_{1},+\infty )\\ \beta_{1}(\frac{1}{p-p_{1}}-\frac{1}{d_{0}}{)}^{2}(p-p_{0}{)}^{\gamma_{1}},\;d\in (d_{0},\;d_{1}]\\ 2\beta_{1}(\frac{1}{p-p_{1}}-\frac{1}{d_{0}}{)}^{2}(p-p_{0}{)}^{2},\;d\in (0,\;d_{0}] \end{array}\right.$$
where $U_{rep}\left(p\right)$ denotes the repulsive potential field function; $\beta_{1}$ denotes the repulsive coefficient; $\gamma_{1}$ denotes the repulsive factor; and $d_{1}$ denotes the distance threshold.

The gravitational function is
(23)$$F_{att}\left(p\right)=-\nabla U_{att}\left(p\right)=\left\{\begin{array}{c} -2\alpha_{1}\left(p-p_{0}\right),\;d\in (0,\;d_{0})\\ -\frac{\alpha_{1}d_{0}\left(p-p_{0}\right)}{\left\|p-p_{0}\right\|},\;d\in (d_{0},+\infty ) \end{array}\right.$$

The repulsion function is
(24)$$F_{rep}(p)=-\nabla U_{rep}\left(p\right)=\left\{\begin{array}{c} 0,\;{d\in (d}_{1},+\infty )\\ F_{1}+F_{2},\;{d\in (d}_{0},\;d_{1})\\ F_{1}+F_{2},\;d\in (0,\;d_{0}) \end{array}\right.$$
where $F_{1}$ and $F_{2}$ are
(25)$$F_{1}(p)=\left\{\begin{array}{c} 2\beta_{1}\left(\frac{1}{p-p_{1}}-\frac{1}{d_{0}}\right)\frac{{\left(p-p_{0}\right)}^{\gamma_{1}}}{{\left(p-p_{1}\right)}^{2}},\;d\in (d_{0},\;d_{1})\\ 2\beta_{1}\left(\frac{1}{p-p_{1}}-\frac{1}{d_{0}}\right)\frac{{\left(p-p_{0}\right)}^{2}}{{\left(p-p_{1}\right)}^{2}},\;d\in (0,\;d_{0}) \end{array}\right.$$
(26)$$F_{2}(p)=\left\{\begin{array}{c} 2\beta_{1}\left(\frac{1}{p-p_{1}}-\frac{1}{d_{0}}\right){\left(p-p_{0}\right)}^{\gamma_{1}-1},\;d\in (d_{0},\;d_{1})\\ 2\beta_{1}\left(\frac{1}{p-p_{1}}-\frac{1}{d_{0}}\right)\left(p-p_{0}\right),\;d\in (0,\;d_{0}) \end{array}\right.$$
where $F_{1}$ and $F_{2}$ denote the two forces of repulsion; here is a segmented design of the repulsion.

The schematic diagram of the extended artificial potential field is shown in Figure 3. “!” denotes the danger zone; “T” denotes the target point.

## 4. Experimentation and Simulation

We utilized MATLAB/Simulink to test the performance of the CISCS algorithm from different angles.

### 4.1. Formation Controller Simulation

In the following, the effectiveness and reliability of the multi-UAV distributed self-organized collaborative intelligent reconnaissance combat strategy (CISCS) proposed are verified through example solutions. The simulation platform is a kind of multi-UAV simulation platform running in MATLAB R2019a/Simulink, as shown in Figure 4, Figure 5 and Figure 6.

In this simulation platform, the UAV body model consists of a UAV power model, a control efficiency model, and a rigid body model, and the UAV controller consists of a position controller and an attitude controller, both of which are PID controllers, and the input of the controller is the desired position, and the output is the PWM control command of the UAV motor, which is inputted into the UAV body model to control the UAV position attitude, and the UAV position attitude is fed back to the controller.

Formation control analysis: in order to conduct formation control performance analysis, we compare the two control methods of adding a formation controller and not adding a formation controller and set up comparison experiments of dynamic flight formation and static flight formation to verify the rapidity and stability of the formation controller. It can be seen that the formation controller can help multi-UAVs form stable flight formations faster.

In Table 2, we give the list of parameters, the selection of which relies on several experiments to obtain the best possible performance optimization.

Firstly, a formation simulation experiment is carried out using one Leader and two Followers; the position of each UAV in 3D space is shown in Table 3; multiple drones are placed in horizontal rows on the ground with a spacing of 1 m. The triangular formation Follower forms a 45° angle with the Leader at a distance of $\sqrt{2}$ m. The circular formation Follower forms an enclosing circle around the Leader with a radius of 2 m. The UAVs in the triangular shape are then simulated in a circular shape. After simulation, it is obtained, as shown in Figure 7, Figure 8, Figure 9 and Figure 10.

Extracting the simulation data, it can be seen that the formation time with the formation controller is significantly shorter than the formation time without the formation controller. Without changing the UAV’s own controller parameter, the formation controller can help the UAV form a triangular static formation faster, compared with the formation without a formation controller, to accelerate the formation by 3.5080 s. In Figure 8b, the UAV spacing is jittered due to the parameter adjustment problem of the formation controller, but the maximum range of the jittering is ±18.79% of the control amount, and the maximum value of the UAV spacing without the formation controller is 2.7663 m, which is 95.64% of over-regulation. It can be concluded that in static three-aircraft formation, the formation controller can form a faster and more stable formation.

In Figure 9a, without adding the formation controller, the two Follower-drones’ paths cannot be overlapped, and in Figure 9b, it can be seen that the distance control of multiple UAVs is unstable when they are in a circular formation around the Leader; after adding the UAV controller, the Follower-drones’ paths in Figure 10a are basically overlapped, and the UAV distance in Figure 10b can be stabilized around 2 m, and the overshooting is 95.64%. Stabilization can be stabilized near 2 m, and the formation time is 16.3720 s, which is larger than the formation time of 15.84 s for static formation, indicating that the formation of dynamic formation is more difficult. Under the condition of not changing the controller parameter of the UAV itself, in the three-aircraft dynamic formation, the formation effect without a formation controller is poor, and the UAV spacing jerks violently; after the addition of the formation controller, there is a small fluctuation in the UAV spacing, but the overall tendency is stable, and the overshooting amount is 27.71%. Therefore, in the three-aircraft dynamic formation, the formation control can help the UAV form a more stable formation.

To verify the scalable performance of this formation controller, formation simulation experiments with five machines are added.

In the five-aircraft static formation control experiment, the comparison between Figure 11a and Figure 12a are obvious; in the presence of the formation controller, the UAV formation is faster and with less error, which accelerates the formation by 3.992 s compared to the formation without the formation controller. In Figure 11b, the formation can be stabilized, but it is obvious that the overshoot is too large compared to Figure 12b, and the formation time is much longer. Although there is some jitter in Figure 12b, it is due to the UAV airframe model, and the range of jitter is small.

In the five-aircraft dynamic formation experiment, there is a large formation error in the Follower-drones in Figure 13a, which may be due to the difficulty of stabilizing the spacing of multiple UAVs during the formation process, which can also be obtained from Figure 13b; after adding the UAV controller, the Follower-drones in Figure 14a are able to form more stable circular formations around Leader, Figure 14b. A stable circular formation and the drone spacing in Figure 14 are obviously much more stable than those in Figure 13, indicating that the formation controller can help the multi-drones form a better formation.

### 4.2. Multi-UAV Cooperative Mission Simulation

In order to verify the effectiveness of CISCS, we call one, three, and five airplanes to perform the same mission and carry out simulation experiments according to the flow of takeoff, formation, formation flight, mission execution, and away from the mission area, and obtain the multi-aircraft mission sequences by CISCS, as shown in Figure 15, Figure 16, Figure 17 and Figure 18.

Task point coordinates are shown in Figure 15 and Table 4.

The mission execution sequence and mission completion time obtained from CISCS are shown in Table 5. From this, it can be seen that there is a reduction of 52.55% and 58.27% for three and five airplanes, respectively, as compared to a single airplane.

## 5. Conclusions

We propose a multi-UAV-distributed self-organized collaborative intelligent reconnaissance combat strategy. In this strategy, a multi-UAV formation controller with time constraints is designed for fast formation, and EAPF is utilized to prevent collisions within the multi-UAV formation, while a distributed improved ant colony algorithm based on Q-Learning is utilized to generate a sequence of tasks executed by multi-UAVs. Simulation experiments show that the multi-UAV finite-time formation controller can help multi-UAVs form stable formations faster and significantly improve the efficiency of task execution by assigning multiple task points to multiple UAVs through the task assignment algorithm, and prevent internal collisions within the UAV formation to guarantee the air safety of UAVs through the Extended Artificial Potential Field (EAPF) method.

CISCS assumes that UAVs are equipped with enough attack munitions and reconnaissance equipment to reconnaissance and attack arbitrary targets, and in the next research, the actual reconnaissance and combat capabilities of UAVs should be considered, increased with dynamic constraints, and multi-dimensional multi-platform collaborative combat issues, such as multi-kinds of UAVs, and ground equipment should be considered.

## Figures and Tables

**Figure 1 sensors-23-07398-f001:**
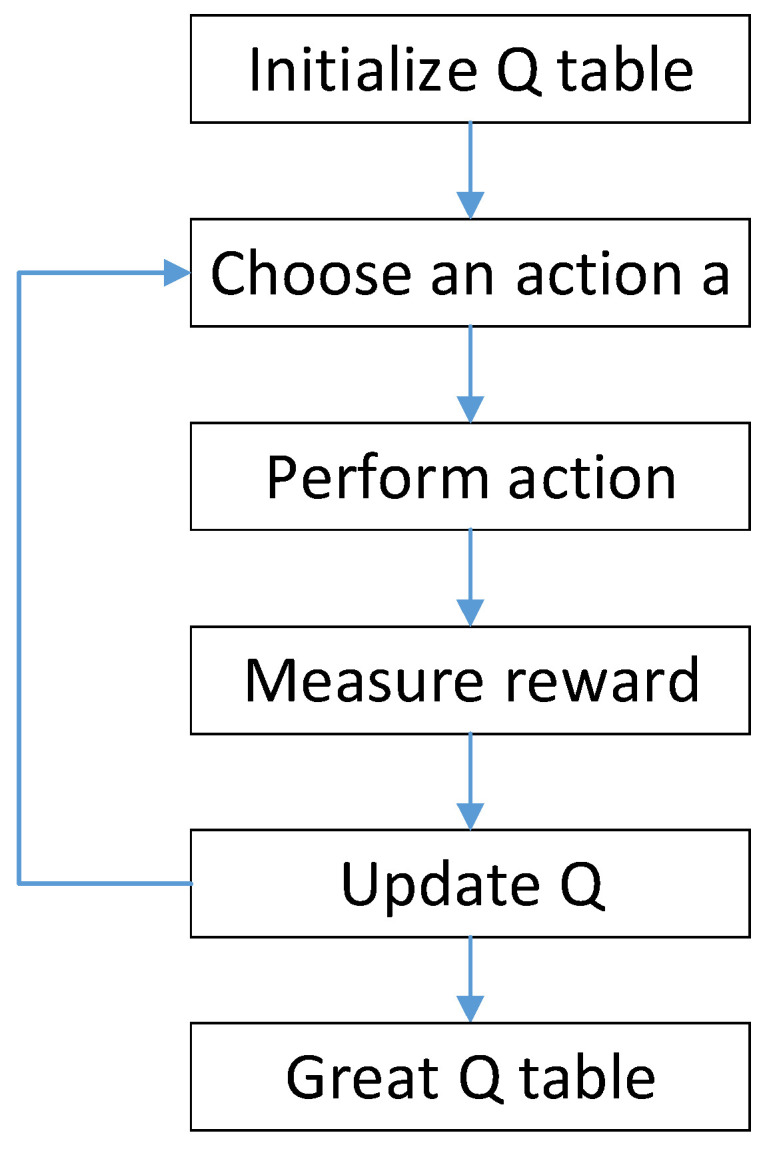
Q-Learning Algorithmic.

**Figure 2 sensors-23-07398-f002:**
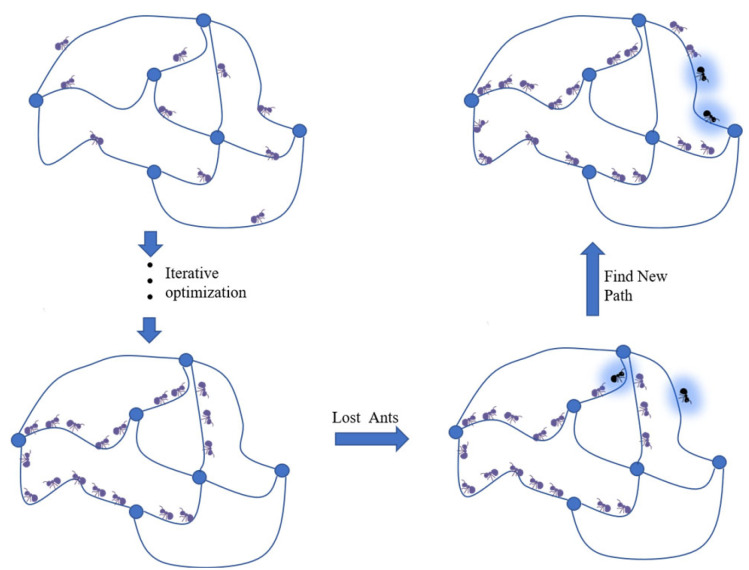
Ant colony lost strategy.

**Figure 3 sensors-23-07398-f003:**
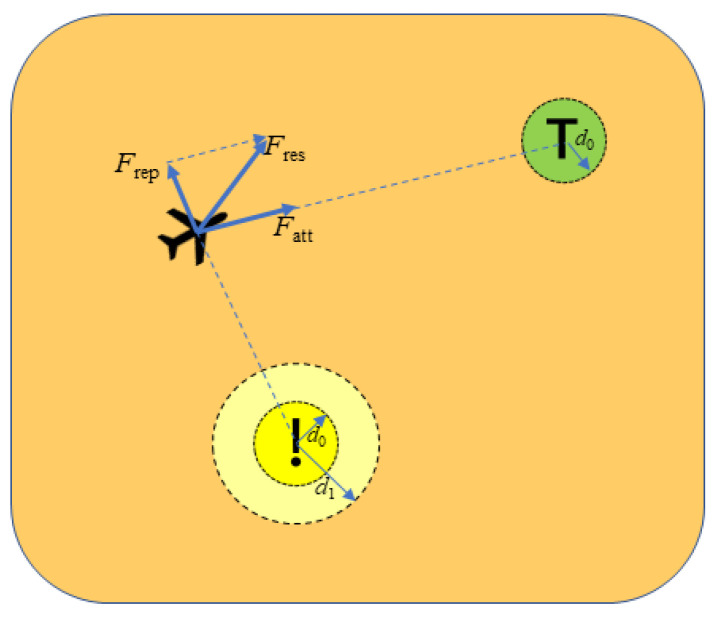
Schematic diagram of EAPF.

**Figure 4 sensors-23-07398-f004:**
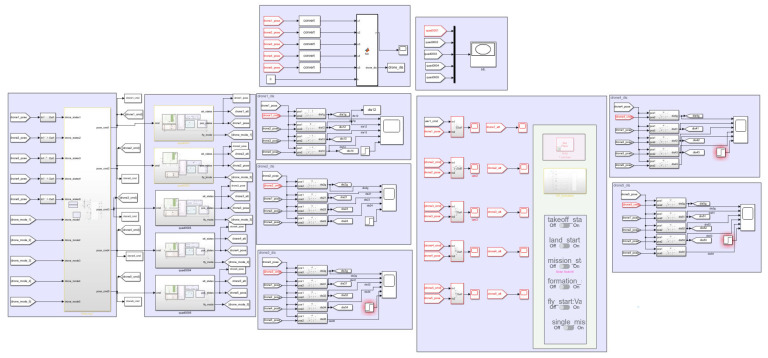
Multi-UAV simulation module.

**Figure 5 sensors-23-07398-f005:**
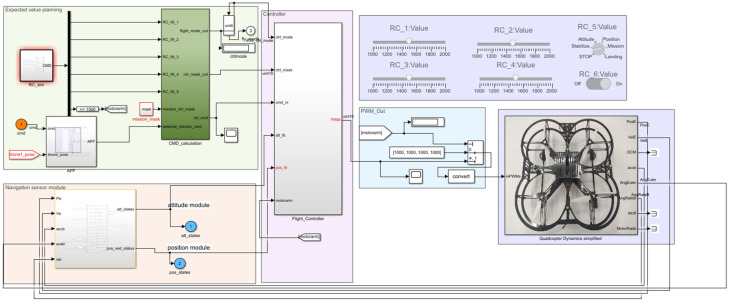
Stand-alone control model.

**Figure 6 sensors-23-07398-f006:**
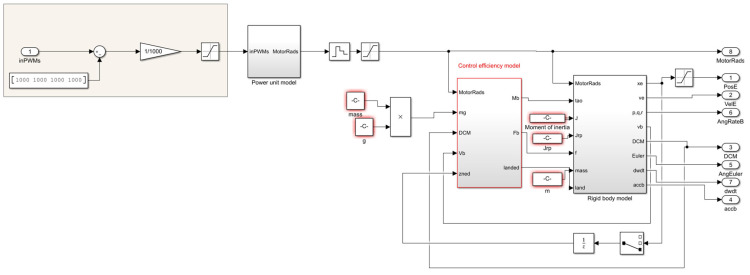
Quadcopter UAV control model.

**Figure 7 sensors-23-07398-f007:**
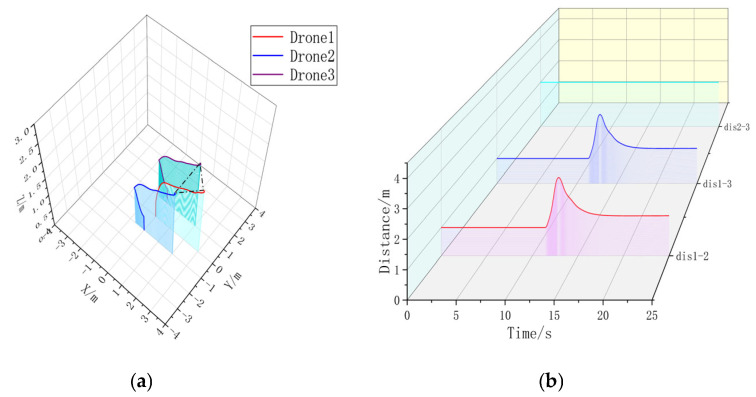
(**a**) A 3D view of the non-formation controller. (**b**) Distance between UAVs without formation controller. Formation time = 15.84 s.

**Figure 8 sensors-23-07398-f008:**
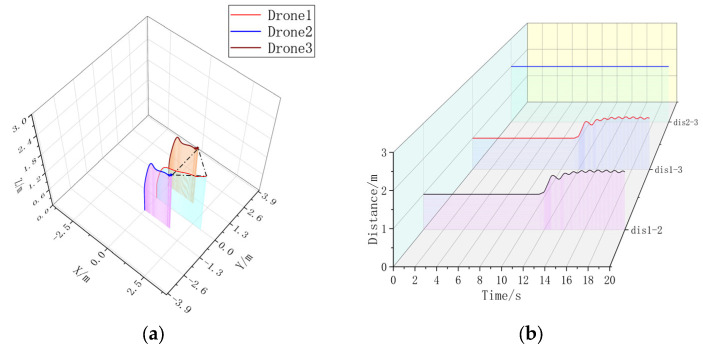
(**a**) A 3D view with formation controller. (**b**) Distance between UAVs with formation controllers. Formation time = 12.3320 s.

**Figure 9 sensors-23-07398-f009:**
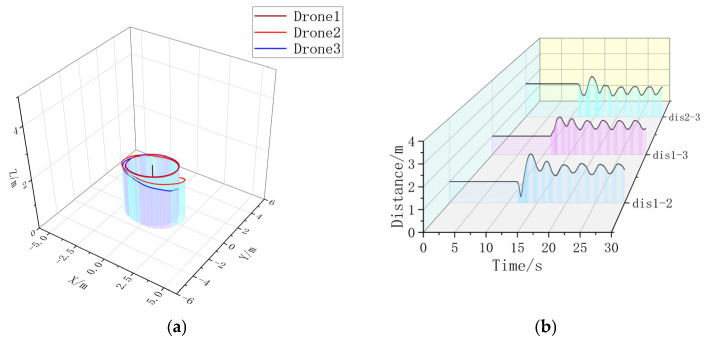
(**a**) A 3D view of the formation-free controller. (**b**) Distance between UAVs without formation controller.

**Figure 10 sensors-23-07398-f010:**
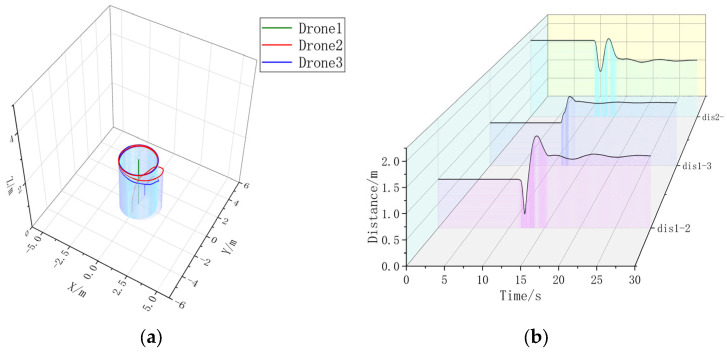
(**a**) A 3D view with formation controller. (**b**) Distance between UAVs with formation controllers. Formation time = 16.3720 s.

**Figure 11 sensors-23-07398-f011:**
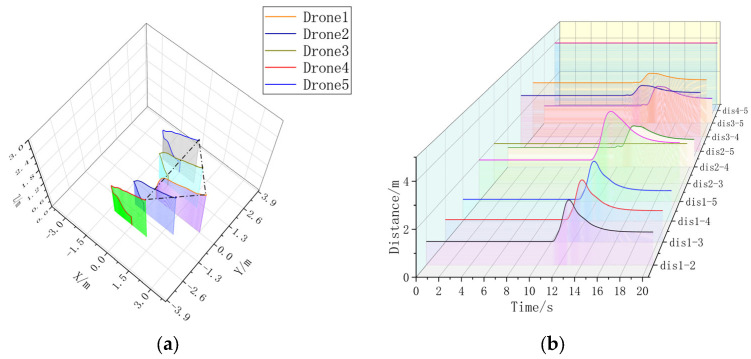
(**a**) A 3D view without formation controller. (**b**) Distance between UAVs without formation controller. Formation time = 17.5360 s.

**Figure 12 sensors-23-07398-f012:**
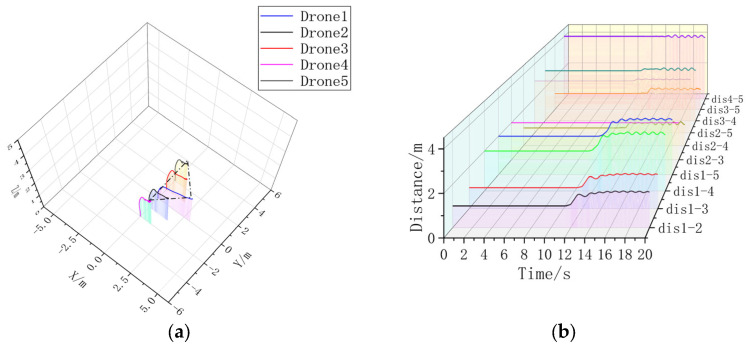
(**a**) A 3D view with formation controller. (**b**) Distance between UAVs with formation controllers. Formation time = 13.5440 s.

**Figure 13 sensors-23-07398-f013:**
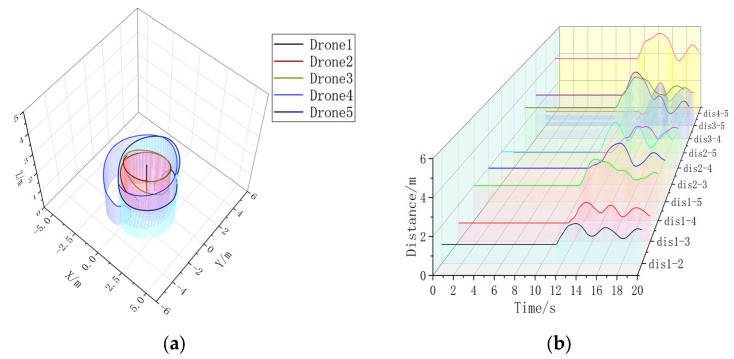
(**a**) A 3D view without formation controller. (**b**) Distance between UAVs without formation controller.

**Figure 14 sensors-23-07398-f014:**
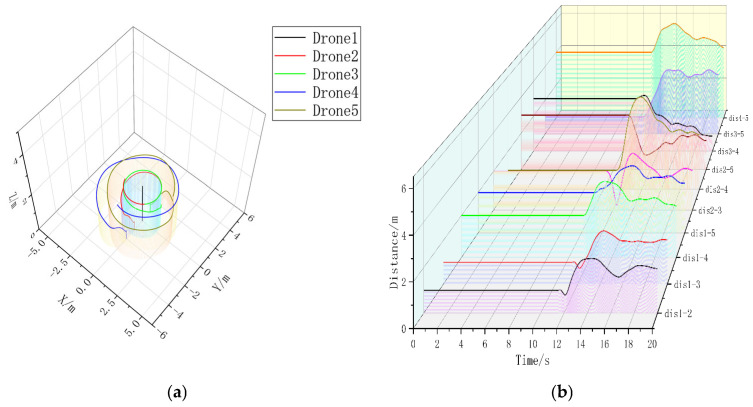
(**a**) A 3D view with formation controller. (**b**) Distance between UAVs with formation controllers.

**Figure 15 sensors-23-07398-f015:**
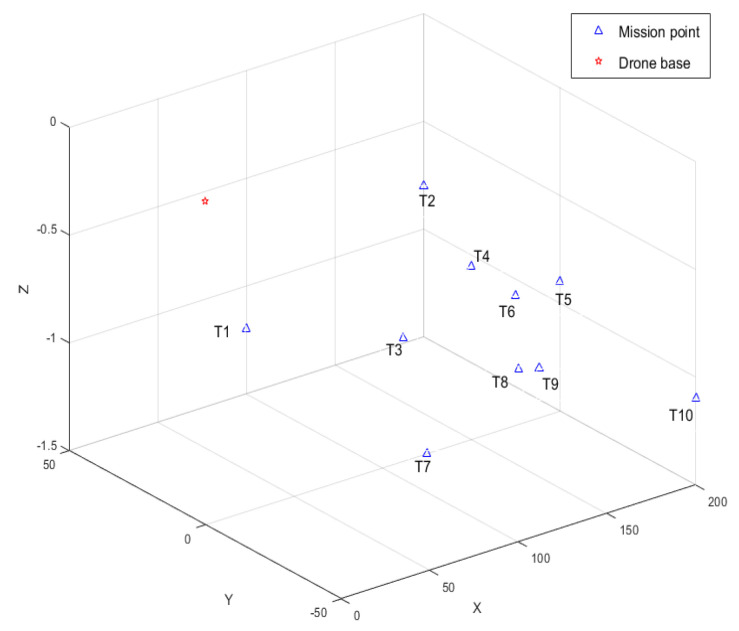
Map of take−off points and mission points.

**Figure 16 sensors-23-07398-f016:**
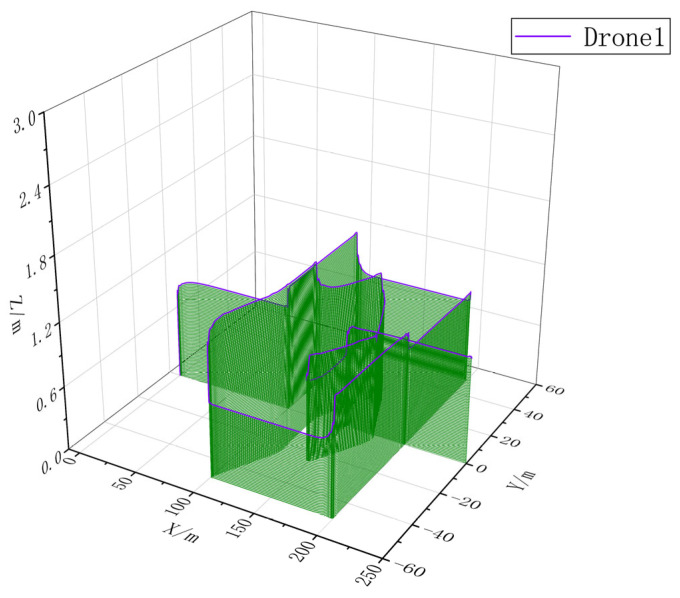
Three–dimensional trajectory of a stand-alone mission flight.

**Figure 17 sensors-23-07398-f017:**
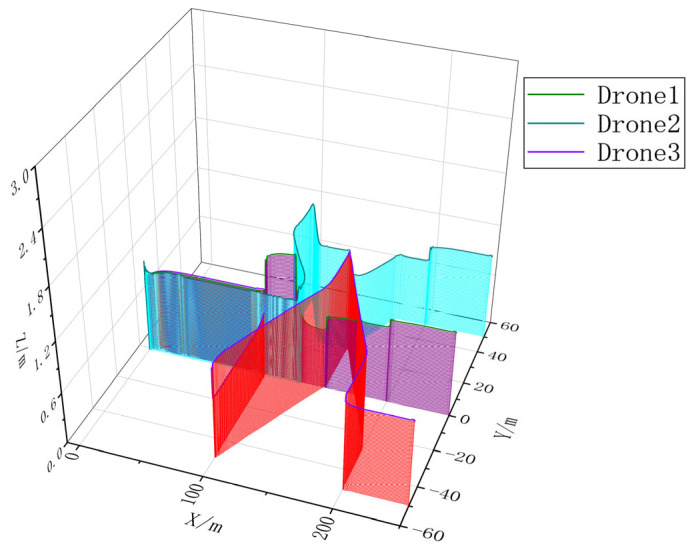
Three-dimensional trajectory of a three-engine mission flight.

**Figure 18 sensors-23-07398-f018:**
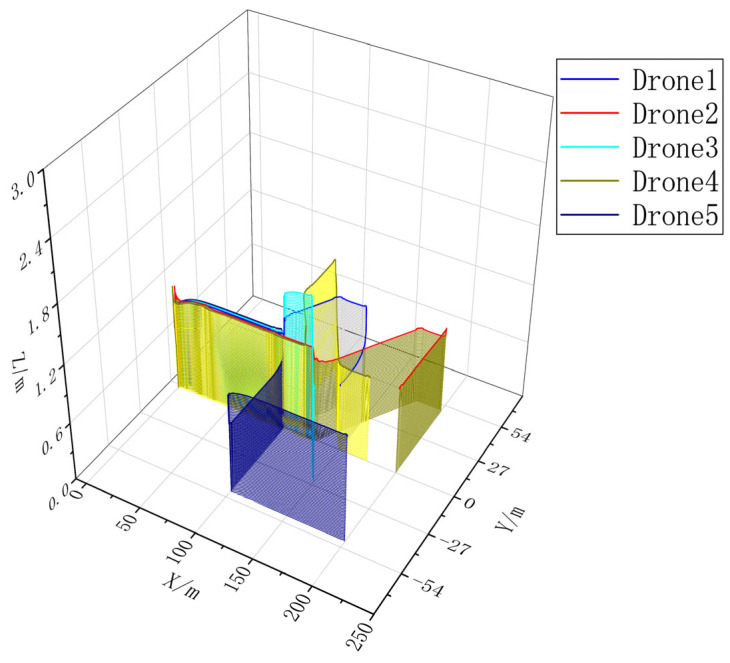
Three-dimensional trajectory of a five-engine mission flight.

**Table 1 sensors-23-07398-t001:** Ant colony update parameters.

	Ant Colony Update Parameters		
	α	β	ρ
Global Search	1.0	3.0	0.15
Local Search	1.25	3.50	0.25
Maintenance Search	1.50	4.0	0.45
Slow convergence	1.75	4.50	0.65
Fast Convergence	2.0	5.0	0.75

**Table 2 sensors-23-07398-t002:** Parameter Setting.

Parameter Meaning	Parameter	Value
Finite time parameter	$t_{w}$	20
Finite time gain	w	1.0
Learning rate	α	0.2
Discount	γ	0.9
Lost rate	τ	0.4
Gravitational coefficient	$\alpha_{1}$	9
Repulsion coefficient	$\beta_{1}$	13
Arrival distance	$d_{0}$	0.1
Distance threshold	$d_{1}$	0.5
repulsive factor	$\gamma_{1}$	1.5

**Table 3 sensors-23-07398-t003:** Initial position of the drone.

UAV	Location
UAV1	(0, 0, 0)
UAV2	(0, 1, 0)
UAV3	(0, −1, 0)
UAV4	(0, 2, 0)
UAV5	(0, −2, 0)

**Table 4 sensors-23-07398-t004:** Task point information.

Number	Location	Number	Location
1	(100, 50, −1.2)	6	(175, 0, −0.9)
2	(200, 50, −0.8)	7	(125, 0, −1.5)
3	(150, 25, −1.2)	8	(100, −50, −0.7)
4	(150, 0, −0.7)	9	(150, −25, −1.0)
5	(200, 0, −0.9)	10	(200, −50, −1.1)

**Table 5 sensors-23-07398-t005:** Sequence of tasks and completion dates.

Number of UAV	Mission Sequence	Time of Mission
1	T1 → T2 → T5 → T10 → T8 → T7 → T3 → T6 → T9 → T4	223.584 s
3	T1 → T3 → T2; T7 → T4 → T6 → T5; T8 → T9 → T10;	106.096 s
5	T3 → T4; T2 → T5; T7 → T9; T1 → T6;T8 → T10	93.312 s

## Data Availability

Not applicable.

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
