# Peer review of "Multi-UAV Collaborative Search and Attack Mission Decision-Making in Unknown Environments"

_sensors, 2023, doi:10.3390/s23177398_

Round 1

Reviewer 1 Report

In this paper, a multi-UAV distributed self-organization Cooperative Intelligence Surveillance and Combat (CISCS) strategy is proposed, which realizes multi-UAV cooperative attack and search task by using the parallel method of control and planning. The method in this paper is reasonable and the results are enriching. However, there are some problems in writing and noun interpretation. Besides there are also some insufficient explanations of control methods and planning methods. The author is requested to correct it.

1.     There is a problem with the symbol format in Section 2 116.

2.     As to whether the Angle in formula 2 includes positive and negative cases, if it does not include directivity, how to explain whether changes other than 180° Angle meet the requirements?

3.     What does it mean that i j in formula (4)?

4.     Can formulas (5) and (6) be combined?

5.     How is the value of \eta in formula (8) calculated?

6.     What is the specific meaning of U_i, J_ri, and J_ai in formula (9), how do they relate to formula (8), and if so, what is the specific meaning of formula (8)?

7.     The finite time formulation in section 3.1 does not seem to be quite the same as formula (10), which is more like discrete time PID control , please explain.

8.     What is the meaning of t_w in formula (11), k_f seems to be proportional gain, Please explain why it was designed this way.

9.     Df should also be a variable related to time T, should it be written as df (t) in formula (10)?

10.  How is formula (15) derived?

11.  Formula (16) does not seem to be relevant to the proof of finite-time stability, if you want to take the finite-time control method, give a specific Lyapunov proof procedure.

12.  For the Q-learning algorithm of reinforcement learning, please give the formula related to the content of this article as a supplementary explanation.

13.  The gravitational and repulsive forces in Figure 2 are associated with formulas (23) and (24), and the meaning of the symbols in Figure 2 is supplemented.

14.  The selection principles of parameters in formula (23) and (24) are also explained.

15.  The parameter list of the control method is not found in section 4, please give it.

16.  Time information is missing in Figure 8 (b).

17.  The explanation of figures (8)-(13) should also be supplemented as far as possible, what results are represented and what conclusions are drawn.

18.  The data in Table 4 does not seem to reach the conclusion of lines 373-376, how to reduce the time? Compared with what method? How the planning method of this article is carried out, please refer to Section 3.2 for details.

19.  This article refers to the actual combat situation, and the following reference can be added to the research of pursuit-evasion game in the actual battlefield:

Ma, X.; Dai, K.; Li, M.; Yu, H.; Shang, W.; Ding, L.; Zhang, H.; Wang, X. Optimal-Damage-Effectiveness Cooperative-Control Strategy for the Pursuit–Evasion Problem with Multiple Guided Missiles. Sensors 2022, 22, 9342. https://doi.org/10.3390/s22239342

Author Response

Please find the attachment below.

Reviewer 2 Report

the article is of interest to researchers who design UAV behavior scenarios in various conditions.

the novelty is the application of a new ant colony algorithm to maintain the formation and behavior of the UAV in the group.

mistakes

1 figure 1 is not representative - should have a more detailed description

2 figure 2 is superfluous - nowhere further the introduced designations are indicated

3 The model is presented in matlab simulink - you should make a formal statement of the problem in advance with given inputs and outputs

4 it is not clear why there are only 5 UAVs in the formation. it is also necessary to clarify whether the results obtained can be used on a larger number of UAVs and under what restrictions (see Table 2). Similarly in Table 3, the end points should be justified.

5 Table 4 shows the mission sequences - where did they come from?

6 Conclusions on the article have insufficient substantiation

7 When describing similar works in the introduction, one should single out those that are very close, and in the final part of the experiment, one should compare with existing solutions.

Round 2

Reviewer 1 Report

Authors have well addressed most of my concerns. It can be accepted after following minor revision

(1) Principles for the value selection of relevant parameters need to be proposed. Even if a trial-and-error approach is adopted, its evaluation principles should also be proposed.
(2) Performance comparision with other literature is necessary to verify the contributions of this manuscript.

Reviewer 2 Report

mistakes:

1 no links to table 2

2 there is no description of the choice of the number of UAVs, parameters and coordinates - tables 2, 3
